

# Interannual-to-multidecadal Hydroclimate Variability and its Sectoral Impacts in northeastern Argentina

Miguel A. Lovino[1,2], Omar V. Müller[1,2], Gabriela V. Müller[1,2], Leandro C. Sgroi[2], Walter E. Baethgen[3]

[1] Consejo Nacional de Investigaciones Científicas y Técnicas (CONICET), Argentina
[2] Centro de Estudios de Variabilidad y Cambio Climático (CEVARCAM), Facultad de Ingeniería y Ciencias Hídricas, Universidad Nacional del Litoral, Santa Fe, Argentina
[3] International Research Institute for Climate and Society, Columbia University, Palisades, United States

*Correspondence to*: Miguel A. Lovino (mlovino@unl.edu.ar)

**Abstract.** This study examines the relation between hydroclimate variability (precipitation, river discharge, temperature) and
water resources, agriculture and human settlements at different time scales in northeastern Argentina. It also discusses the impacts on these productive and socio-economic sectors. The leading patterns of variability, their nonlinear trends, and cycles are identified by means of a Principal Component Analysis (PCA) complemented with a Singular Spectrum Analysis (SSA). Interannual hydroclimatic variability centres on two broad frequency bands: one of 2.5-6.5 years corresponding to El Niño Southern Oscillation (ENSO) periodicities and the second of about 9 years. Interdecadal variability is characterized by low-
frequency trends and multidecadal oscillations that have induced a transition to wetter and warmer climate starting in the mid-twentieth century. The hydroclimate variability at all time scales had significant sectoral impacts. Frequent wet events between 1970 and 2005 favoured floods that affected agricultural and livestock productivity and forced population displacements. On the other hand, agricultural droughts produced soil moisture deficits affecting crops at critical growth periods. Hydrological droughts affected surface water resources causing water and food scarcity and stressed the capacity for hydropower generation.
Lastly, increases in minimum temperature reduced wheat and barley yields.

## 1 Introduction

Hydroclimate variability affects natural and human systems worldwide. Climate-related extremes such as droughts, floods, and heat waves alter ecosystems, disrupt food production and water supply, damage human settlements and cause morbidity and mortality (Field et al., 2014). Examples around the world include annual losses of about \$6–8 billion to the U.S. economy
due to drought (Carter et al., 2008), and the extraordinarily severe heat wave over western and central Europe in the summer of 2003 that produced excess deaths of about 35,000 (Kosatsky, 2005) with estimated economic losses for the agriculture sector in the European Union at € 13 billion (Sénat, 2004).

The impacts of hydroclimate variability are more evident in regions where population and the productive sectors are vulnerable to climate hazards. Southeastern South America (SESA) is one such region as documented by Magrin et al. (2014). Frequent
flooding impacted large populated areas over SESA (Andrade and Scarpatti, 2007; Barros et al., 2008a). The extraordinary





flood along the Paraná River in 1983 produced economic losses of more than \$1 billion and forced the evacuation of at least 100,000 people (Krepper and Zucarelli, 2010). The extended and persistent drought of 2008/2009 caused losses of about 40% of grain production in Argentina and greatly affected the hydroelectric sector over SESA (Bidegain, 2009). In this context, it is necessary a better understanding of regional hydroclimate and its sectoral impacts to increase the resilience of the affected

populations by providing adequate information that will facilitate decision-making processes.

Hydroclimate in SESA varies on interannual to multidecadal time scales. On interannual time scales, El Niño Southern Oscillation (ENSO) is the major source of hydroclimate variability (Garreaud et al., 2009). El Niño conditions can cause increased severe precipitation and streamflow while La Niña events may favour droughts (Grimm et al., 2000; Camilloni and Barros, 2003; Penalba and Rivera, 2016). Precipitation and Paraná River flow present oscillations of 3 to 6 years linked to the

ENSO and a near-decadal cycle related with the North Atlantic Oscillation (NAO) (Robertson and Mechoso, 1998; Krepper and García, 2004; Antico et al., 2014). Regarding temperature, the interannual variability modes of extreme temperature frequencies concentrated on a 2-4 years band and an 8-years signal mostly associated with the Southern Annular Mode (Barrucand et al., 2008; Loikith et al., 2017)

On decadal-to-multidecadal time scales, the Pacific decadal variability (PDV) also modulates SESA hydroclimate in the same

way as ENSO, i.e., positive SST anomalies might favour wetter conditions while negative SST anomalies might favour drier conditions (Andreoli and Kayano, 2005). Further, there are other sources of hydroclimate variability such as the Atlantic Ocean and the South Atlantic Convergence Zone (SACZ) (Seager et al., 2010; Mo and Berbery, 2011; Grimm and Saboia, 2015; Grimm et al., 2016). For example, the Paraná streamflow is dominated by a near bi-decadal oscillation related to the SACZ and a multidecadal cycle forced by PDV (Antico et al., 2014).

Hydroclimate trends have been observed over SESA. Precipitation and temperature extreme events registered long-term increases since the 1960s as well as Paraná River flow after mid-1970 (Haylock et al., 2006; Seneviratne et al., 2012; Cavalcanti et al., 2015; Carril et al., 2016; Scardilli et al., 2017). Huang et al. (2005) and Jaques-Coper and Garreaud (2015) have suggested that those changes were influenced by interdecadal variability in tropical Pacific SSTs. However, the wet trend in precipitation could also be favoured by a multidecadal cooling in the tropical Atlantic Ocean (Seager et al., 2010; Barreiro et

al., 2014).

Summarizing, SESA hydroclimate is driven by several modes of variability forced by SST patterns at different time scales. The interannual modes mainly associated with the ENSO and the NAO prove to be the most significant as they explain almost 60% of the total precipitation variance over the 20th century (Baethgen and Goddard, 2013). The multidecadal modes including long-term trends are mainly driven by the Pacific and the Atlantic SSTs.

This study has three main purposes: first, to reassess the joint variability from interannual to multidecadal scales of precipitation, river flow and maximum and minimum temperatures over northeastern Argentina. Second, to review and discuss the links between those regional hydroclimatic variables and global SSTs forcing. Last, to assess the impacts of hydroclimate variability and trends on water resources, agriculture and human settlements. The structure of the article follows: section 2 introduces the study region, dataset and methods. Section 3 describes the leading pattern of global SST variability. Section 4





discusses hydroclimatic variability at interannual-to-multidecadal time scales and its potential links with global SST leading modes. Section 5 addresses the main sectoral impacts of regional hydroclimatic variability and section 6 offers the final remarks.

## 2 Methodology

### 2.1 Study region

The study region is located on southeastern South America covering La Plata Basin (Fig. 1a), which has four major sub-basins: Uruguay, Paraguay, Mid-Upper and Lower Paraná (see Fig. 1b). The Paraguay and Mid-Upper Paraná sub-basins contribute with a high percentage of the Paraná River flow. The Lower Paraná River extends over a flat plain, where high discharges easily cause severe floods (Coronel and Menéndez, 2006). A large portion of the Lower Paraná sub-basin is covered by the study region in northeastern Argentina (NEA), delimited by 36°-26°S and 65°°-58°W (red box in Fig. 1b). This region is key for the socio-economic development of the country and the continent. It concentrates most of the country's population, has a complex system of water resources management and assembles eighty percent of the cereal production in Argentina (Ministry of Agroindustry, 2017).

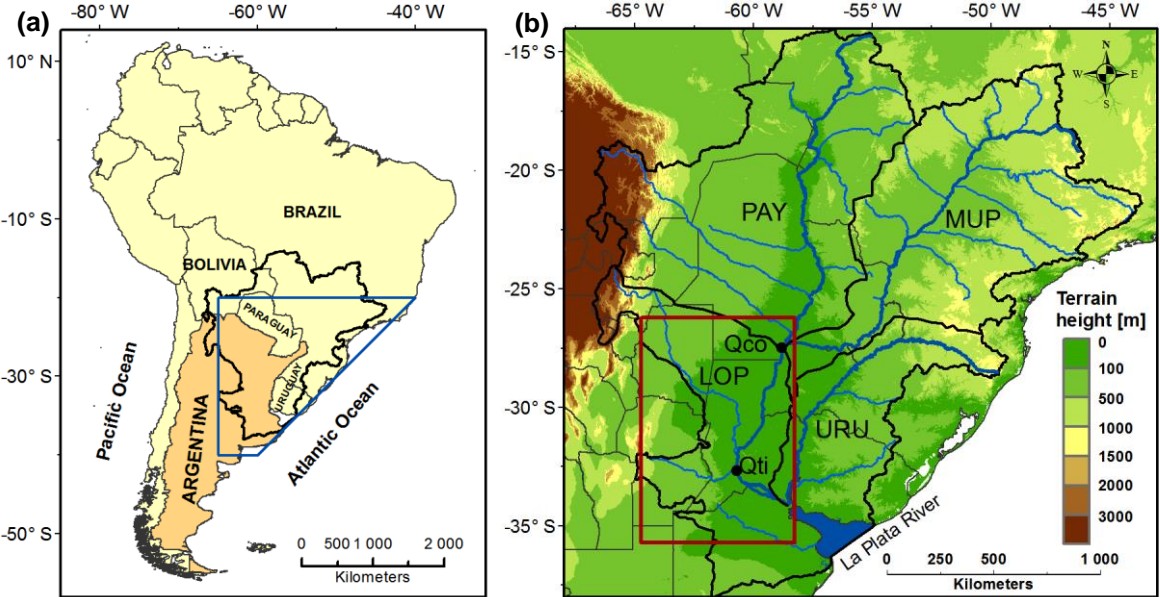

**Figure 1: (a) South America and the domains of the La Plata Basin (black contour) and southeastern South America (blue contour). (b) Topography map of La Plata Basin and its sub-basins, Paraguay (PAY), Mid-Upper Paraná (MUP), Lower Paraná (LOP) and Uruguay (URU). Gauging station of the Parana River streamflow in Corrientes (Qco) and Timbúes (Qti) are highlighted. The study area in northeastern Argentina is highlighted with a red rectangle.**



## 2.1 Datasets

Regional hydroclimate variables are available from the beginning of the 20th century to about the first decade of the 21st century. The observed precipitation dataset used is the Global Precipitation Climatology Centre dataset version 7 (GPCC v7, Schneider et al., 2015). GPCC v7 consist of monthly gridded precipitation dataset with a 0.5º x 0.5º degree spacing extending from January 1901 to December 2013. According to Lovino (2015), GPCC v7 has lower biases and fits extreme fluctuations better than other precipitation datasets over the study region. While there are areas in SESA with poor rain gauge coverage, northeastern Argentina has a reasonable station distribution for the whole period (Barreiro et al., 2014).

Monthly streamflow along the Paraná River is measured at two locations: the Corrientes and Timbúes gauging stations (Fig. 1b). These observed streamflow data might include variations due to anthropogenic influence, particularly damming. The Corrientes station represents most of the Parana basin discharge since the main tributaries are upstream (Camilloni and Barros, 2000; Berbery and Barros, 2002). Corrientes station data covers from January 1904 to August 2014 with no missing values. Timbúes is further downstream and reflects precipitation variability in the Lower Paraná basin. Timbúes station data extends from September 1905 to August 2014 with only 4 missing values that were linearly interpolated to have a continuous series. All data were obtained from the Integrated Hydrological Data Base of the National Water Information System of Argentina (NWIS, 2016).

Monthly time series of daily maximum and minimum temperatures used are obtained from the Climatic Research Unit (CRU TS 3.23, Harris et al., 2014). The CRU dataset has a resolution of 0.5° x 0.5° degrees and covers the period 1901-2014. CRU TS 3.23 also fits well observed mean and extremes temperatures throughout NEA (see Lovino, 2015). Lastly, monthly SST data are those from the Extended Reconstructed Sea Surface Temperature Version 4 (ERSSTv4, Huang et al., 2015; Liu et al., 2015). ERSSTv4 dataset is available on a global 2º x 2º grid since January 1854.

Historical journalistic information was collected to document the impacts of the key cases of extreme events related to hydroclimatic variability. We also compile and analyse the scientific literature regarding the potential impacts of droughts, floods and heat waves over southeast South America. The changes and variability of the regional precipitation, flows and temperature are identified and then used as input variables to determine their joint sectoral impacts.

## 2.3 Statistical approach

The present analysis focuses on precipitation, streamflow and temperature (minimum and maximum) of the NEA, considering SSTs as the main external forcing. For these four variables, monthly mean anomalies are computed by removing the corresponding mean annual cycle. The variables are also filtered applying low-pass Lanczos filters (Duchon, 1979) with cut-off periods at 18 and 120 months to emphasize the interannual and low frequency behaviour respectively.

The Principal Component Analysis (PCA, Von Storch and Zwiers, 1999; Wilks, 2006) is applied to extract the leading global SST patterns and to evaluate the spatiotemporal variability of the precipitation fields. PCA is used in S-mode, therefore, the principal components (PCs) are time series and the eigenvectors or empirical orthogonal functions (EOFs) are spatial patterns



that vary in time according PCs. Although rotation of the EOFs has advantages, like avoiding the influences of different processes in a single PCA pattern, we opted not to do it because in our case it led to patterns overly localized in space, i.e., several EOFs concentrate information of the spatial patterns in small regions hindering their interpretation (not shown; see Deser et al., 2010).

A Singular Spectrum Analysis (SSA, Ghil et al., 2001) is then used to study the temporal variability of (1) leading PCs of precipitation and SST, (2) streamflow time series and (3) temperature anomalies time series. SSA determines the structures of deterministic signals as nonlinear trends and quasi-oscillatory modes. In the SSA method, the window length M should not exceed $M = N / 3$, where N is the length of the time series, to adequately represent quasi-cycles between M/5 and M (Von Storch and Navarra, 1995). For decadal-to-multidecadal variability, we considered a window length that retains variability of

the 120-month low pass filtered series between 10 and 30 years (M=360 months or 30 years; M/5 < 120 months or 10 years of the low-pass filter). For Interannual variability, we considered a windows length that retain variability of the 18-month low pass filtered series between 2 and 10 years (M = 120 months or 10 years and M/5 = 24 months or 2 years).

Sectoral impacts of hydroclimatic variability are evaluated at different time scales. First, we identify the key characteristic of the regional hydroclimate variability on interannual-to-multidecadal time scales and the influence of global SSTs forcing.

Changes in precipitation, streamflow and temperature are studied by nonlinear trends. Then, we address how the main properties of the variability (including extreme events) and changes affect water resources, agriculture, and human settlements. Finally, we discuss the sectoral impacts attributed to hydroclimate variability analysing some key cases and reviewing the information available in the scientific literature.

## 3 Leading patterns of global SST variability

The patterns of global SSTs have been widely studied and evaluated in different time scales and periods (e.g., Schubert et al., 2009; Deser et al., 2010; Messié and Chavez, 2011). Other studies have discussed the mechanisms by which the SST patterns affect SESA hydroclimate (e.g., Grimm et al., 2000 and Seager et al., 2010 among many others). Our main interest is to discuss the leading modes of global SST variability and examine their links to significant modes of hydroclimatic variables in NEA as a necessary step to understand their sectoral impacts. Figure 2 presents the three leading EOFs and their corresponding PCs

of 18- and 120-month low-pass filtered monthly mean global SST fields. The three main patterns explain more than 50% and 70% of the total variance of global 18- and 120-month filtered SSTs fields, respectively. The global SST patterns are very similar to those obtained by Schubert et al. (2009) using annual mean SSTs. Slight differences result from using monthly data instead of yearly data, and because the rotation was not done. Table 1 presents the dominant modes of interannual and low-frequency SST variability for each PC.





**Figure 2: First three leading global pattern of 18- and 120-month low-pass filtered monthly mean SST between jan-1901 and sep-2012. Patterns are described by the spatial loadings (EOFs) and their associated PCs. The percentage of variance explained by each pattern is given in brackets. Partial reconstructions at interannual time scales are plotted in PCs of the left column (IA Rec). Partial reconstructions at low-frequency time scales (LF Reconstruction) are plotted in PCs (lf) of the right column. The partial reconstructions include all variability modes shown in Table 1. Panels (j) and (h) also show the multidecadal oscillations.**



| Time Scale | Sea Surface Temperature | | |
|---|---|---|---|
| | PC1 (Var) | PC2 (Var) | PC3 (Var) |
| **Interannual** (18-month low-pass filtered time series) | 3.5 (3) | 2.8 (10) | 4.3 (3) |
| | | 3.5 (17) | |
| | | 5.7 (32) | |
| | 9.3 (6) | | |
| | PC1 (lf) (Var) | PC2 (lf) (Var) | PC3 (lf) (Var) |
| **Decadal-to-multidecadal** (120-month low-pass filtered time series) | 20-26 (5) | 13-15 (10) | 14 (8) |
| | | | 20-27 (20) |
| | Trend (87) | Multidecadal oscillation (75) | Multidecadal oscillation (67) |

**Table 1. Trends and dominant periodicities obtained from applying SSA to the first three PCs and PCs (lf) of the 18- and 120-month low-pass filtered SST global patterns, respectively. The analysis is performed on interannual timescales (between 2 and 10 years) and lower frequency or decadal-to-multidecadal timescales (higher than 10 years). The percentage of variance explained by each mode is given in brackets.**

The first pattern of 18-month low-pass filtered global SST (Fig. 2a) displays positive loadings in most of the world ocean. The first EOF presents the highest positive loadings in the South Atlantic and the Indian Ocean, congruent with the results of Schubert et al. (2009). Also, high positive loadings are shown in the central Pacific Ocean. The first PC (Fig. 2b) is driven by a low-frequency nonlinear trend, but interannual time scales also influence PC1 with periodicities of roughly 9 and 3 years that explain 9% of its variance (see Table 1 and the interannual reconstruction in Fig. 2b).

The second pattern (Fig. 2c) reveals a pan-Pacific pattern resembling features of the ENSO. The EOF2 displays positive loadings from the central to the eastern Pacific and negative values in the North and South Pacific. The spectral decomposition of the corresponding PC2 presented in Table 1 shows dominant modes at interannual time scales between 2.8 and 5.7 years. The PC2 shown in Fig. 2d has correlations of about 0.9 with indices of ENSO evolution.

The third pattern (Fig. 2e) presents positive loadings in the North Atlantic Ocean and negative loadings south of 40°-50°S, with centres in the South Atlantic and Indian Oceans. The PC3 (Fig. 4f) is mostly driven by a multidecadal oscillation. The interannual variability exerts a slight effect through a cycle of approximately 4 years that represents only 3% of the PC1 variance (see Table 1 and the interannual reconstruction in Fig. 4f).

Decadal-to-multidecadal SST variability is characterized by the 120-month low-pass filtered SST patterns (right column Fig. 2). The first pattern (Fig. 2g) displays the highest positive loading in the south Atlantic and Indian Ocean but not in the central Pacific Ocean as the EOF1 in Fig. 2a. The PC1 (lf) (Fig. 2h) exhibits a nonlinear trend that explains more than 85% of its variance. The trend denotes an increase in global SSTs over the whole period, reaching positive anomalies since the 1960s.

The second pattern (Fig. 2i) resembles features of the Pacific Decadal Oscillation. Also weakly positive loadings are shown in the central and southwest Pacific Ocean and negative loadings towards the North Atlantic Ocean. PC2 (lf) presents two cycles,





one close to 15 years and a multidecadal oscillation of very low frequency (Table 1). Although the multidecadal oscillation present periodicities out of the range that can be estimated by SSA, the spectral analysis allow us inferring periodicities of around 70 years. These decadal-to-multidecadal periodicities exhibit in Fig. 2j are positively correlated to the ones of the Decadal and Interdecadal Pacific Oscillations (as identified by Mantua and Hare, 2002 and Henley et al., 2015 respectively).

The third pattern (Fig. 2k) presents positive loadings in the North Atlantic Ocean that resemble the Atlantic Multidecadal Oscillation pattern (Enfield et al., 2001). The PC3 (lf) (Fig. 4l) shows a multidecadal oscillation that account for near 70% of the PC (lf) variance (Table 1). The dominant frequency of the multidecadal oscillation is out of range but it seems to be about 70 years. Schlesinger and Ramankutty (1994) among others have reported a similar oscillation with the same frequency in the North Atlantic. On the other hand, variability in the 10-30 years range represents about 30% of the total PC3 (lf) variance.

Different authors have found similar frequencies in the South and North Atlantic Oceans (Venegas et al., 1998; Moron et al., 2000).

## 4 Regional hydroclimate variability and its links to global SSTs forcing

The precipitation, streamflow, and temperature over NEA exhibit spectral peaks on interannual and decadal-to-multidecadal time scales that are summarized in Table 2. On interannual time scales, hydroclimate variability centres on two bands: one

with frequencies between 2.5 and 6.5 years and the other with periodicities close to 9 years. For each variable, interannual modes represent more than 50% of their temporal variability; except for minimum temperature, which modes explain 30% of its variance. On decadal-to-multidecadal time scales, nonlinear trends and multidecadal oscillations account for most of the variability, while the interdecadal modes (about 11-25 years) have a lesser effect.

| Time scale | PC1 Pr (Var) | PC2 Pr (Var) | Streamflow (Var) | Tmx (Var) | Tmn (Var) |
|---|---|---|---|---|---|
| **Interannual** (18-m low-pass filtered time series) | 2.4 (17) | 2.5 (15) | 2.4*(7) | 2.4 (13) | |
| | 4 (14) | 3.4 (16) | 3.7 (18) | 3.5 (17) | 3.5 (17) |
| | 6.5 (24) | | 5.8 (15) | | 6.5 (14) |
| | | 9 (19) | 8.8 (27) | 8.8 (26) | |
| **Decadal-to-multidecadal** (120-m low-pass filtered time series) | 15 (4) | 11.5 (25) | | | |
| | 20 (9) | | 18-24 (20) | 24 (34) | 19 (24) |
| | Trend (60) | Multidecadal oscillation (58) | Multidecadal oscillation (62) | Trend (21) | Trend (55) |

**Table 2. Leading frequencies of regional precipitation patterns, monthly mean streamflow of the Paraná River, and area-averaged maximum and minimum temperature (Tmx and Tmn). Precipitation is represented by the two leading patterns of the PCA for both 18- and 120-month filtered fields. All streamflow modes were noted in Corrientes and Timbúes gauging stations, except the near-biannual cycle (*) that is only shown in Timbúes. The percentage of variance explained by each mode is given in parenthesis.**



## 4.1 Interannual modes

Figures 3a and 3b show the spatial pattern and temporal evolution of the first principal component of precipitation. The pattern is mainly located towards the centre-east of NEA with variability within the higher interannual band (2.5-6.5 years). Fig. 3c shows that the bands of variability of the precipitation correspond to the frequencies of the ENSO-like SST pattern. This relationship is well known and has been studied by several previous works (e.g., Grimm et al., 2000; Paegle and Mo, 2002; Garreaud et al., 2009). Our results show that all the main frequencies of the ENSO (i.e., 6, 4, and 2.5 years) are also replicated in the regional precipitation. The 6-year frequencies present the highest correlation values (r~0.61) between ENSO and regional precipitation, while the other frequencies have correlations of about 0.3. The second leading pattern of precipitation (not shown, 11 % of the total variance) is located toward the northwest of NEA. The PC2 presents cycles with periodicities of near 3 years and within the lower interannual band of 9 years (see Table 2).

Interannual variability of precipitation has a close relation with extreme events (see the dots in Fig. 3b). In agreement with Grimm et al. (2000) and Grimm and Tedeschi (2009), Figs. 3b and 3c show that some of the largest extreme precipitation events occurred during extreme ENSO years. Strong El Niño events favoured wet extreme events, such as those in 1914, 1972-1973 and 1997-1998, when SSTs and precipitation cycles were mostly in phase and with high amplitudes. Also, severe droughts were favoured by strong La Niña events, such as those in 1916-1917 and 1988-1989. Again, during those years all precipitation and ENSO cycles were in-phase with large negative anomalies. However, the ENSO phenomenon by itself is not enough to explain the intensity of the droughts and wet events. First, extreme precipitation anomalies were recorded during the 2002-2003 moderate El Niño event, and conversely, the 2008-2009 intense drought occurred under moderate La Niña conditions. Studies have shown that other ocean forcing, as the SST over the North Tropical Atlantic Ocean or the South Atlantic Convergence Zone can combine with ENSO to intensify the wet or dry signals in precipitation, leading to extreme events (Robertson and Mechoso, 2000; Seager et al., 2010; Mo and Berbery, 2011). Second, the precipitation for the 2.5- to 4-yr periods exhibits a large increase in amplitude after 2000 as noted in mid-panels in Fig. 3c. Yet, ENSO-like SST cycles were weakened for the same period. This difference could be explained by an increase in heavy rainfall resulting from greater atmospheric instability and water vapour content (see e.g. Re and Barros, 2009; Penalba and Robledo, 2010). Moreover, these intense precipitation events are strongly influenced by cycles between 2 and 4 years (Lovino et al., 2018).



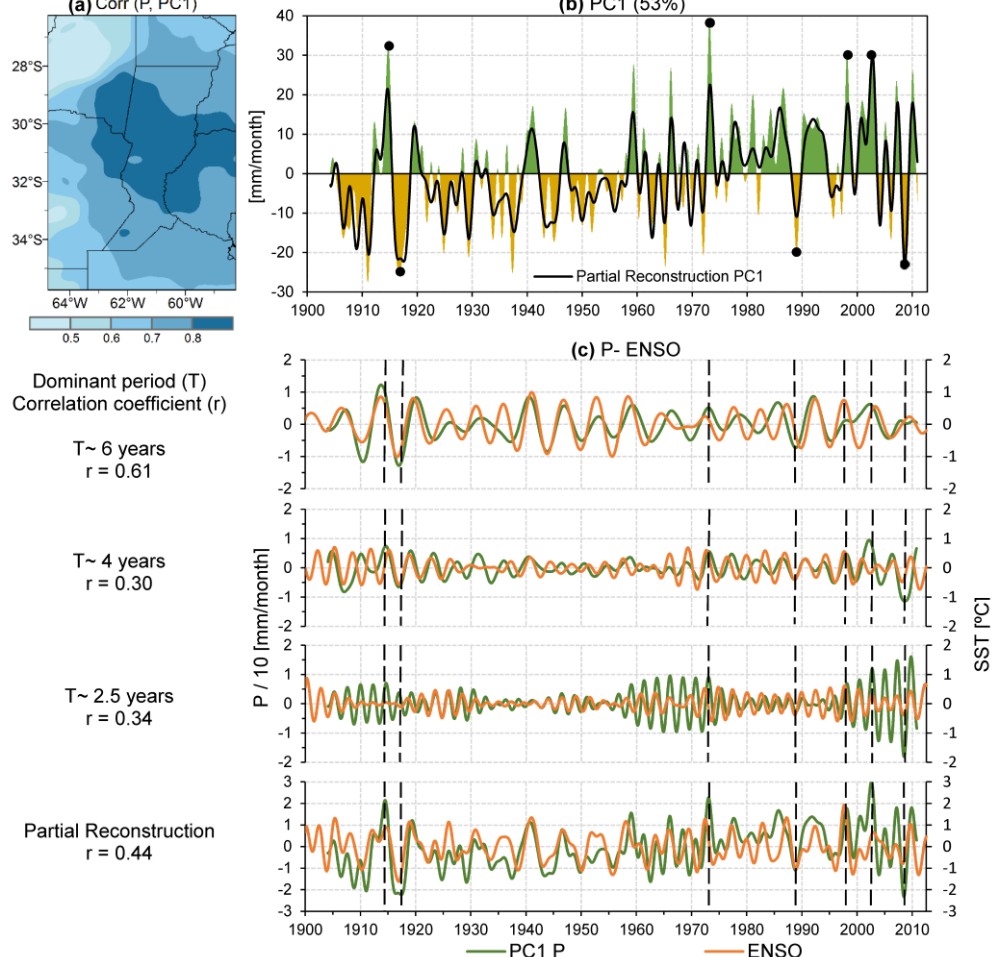

**Figure 3: First leading pattern of the 18-m low-pass filtered precipitation anomalies. Panel (a) shows the spatial distribution of the correlation between PC1 (in panel b) and each precipitation time series at a single grid point. The percentage of explained variance is given in brackets. Panel (c) compares the time series of the dominant cycles of the PC1 of SSTs related to the ENSO pattern and regional precipitation at interannual time scales. Precipitation modes have been scaled to facilitate their interpretation. Vertical dashed lines coincide with months of maximum wet anomalies and the severest droughts (dots in Panel b).**

The streamflow of the Low Paraná River flow is characterized by the flow at the Timbúes station (Fig. 4a). More than 65% of the 18-m filtered series variance is explained by the modes in the 2.5-9 year scales (see Table 2, represented by the IA reconstruction in Fig. 4a). These periodicities are consistent with those reported by García and Mechoso (2005), Krepper et al. (2008) and Antico et al. (2014). As in precipitation, streamflow modes between 2.5 and near 6 years correspond to the ENSO-range SST periodicities (Fig. 4b), in agreement with Robertson and Mechoso (1998) and Antico et al. (2014). We find that the 4-year cycles of ENSO and streamflow reach the largest correlation (r~0.55), as they are in-phase for almost the entire period (mid panel Fig. 4b). Cycles of roughly 2.5 years do not exhibit a strong correlation for the earlier period but become in-phase mainly starting around 1980 (bottom panel Fig. 4b). Interestingly, the 2.5-year mode is noted in Timbúes station time series



but not in Corrientes station (not shown), suggesting a local contribution of NEA precipitation rates to the Low Paraná river streamflow.

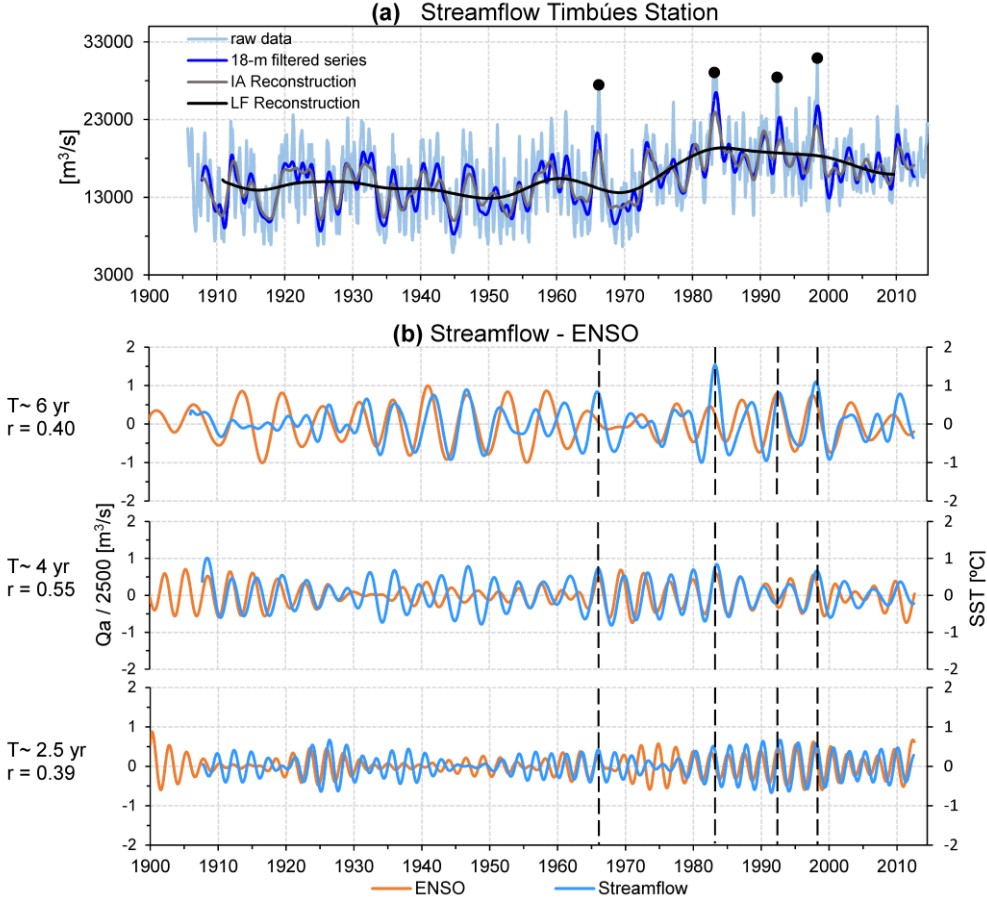

**Figure 4: (a) Monthly mean streamflow time series for the Paraná River at Timbúes station and its partial reconstruction using all significant modes of interannual (IA Reconstruction) and decadal-to-multidecadal variability (LF Reconstruction) presented in Table 2. (b) Time series of the dominant modes of Paraná River streamflow anomalies and the ENSO pattern on interannual time scales. Streamflow modes have been scaled to facilitate their interpretation. Dominant periods (T) of the leading modes and Pearson correlation coefficients (r) are given to the left of the figure. Vertical dashed lines coincide with extreme flood events in 1966, 1983, 1992 and 1998.**

Floods in the Paraná River are highly influenced by the interannual variability (see dots in Figs. 4a). In agreement with Antico et al. (2016), the largest floods that occurred in 1966, 1983, 1992 and 1998 (dots in Fig. 4a) are echoed with peaks for the different interannual modes of streamflow and ENSO-like SSTs (Fig. 4b). These results suggest that all the extraordinary floods of the Paraná River occurred during strong or very strong El Niño events (see also Camilloni and Barros, 2003). As state above, the upper Paraná basin provides the largest amount of water to the Paraná River flow. Precipitation rates over NEA account for about 5% of the total Paraná River flow (Giacosa et al., 2000). Thus, the lower Paraná floods are a direct consequence of excess precipitation in the Upper Paraná Basin that has a closely link with extreme El Niño events (e.g., Camilloni and Barros, 2003; Krepper et al., 2008).



Figures 5a and 5c present the interannual variability of maximum and minimum temperature, respectively. Table 2 shows that Tmx exhibit frequencies close to 3 years and of roughly 9 years while Tmn presents cycles between 3 and 6.5 years within the higher interannual band. The interannual modes strongly influence maximum temperature, explaining almost 60% of its variability (see table 2 and the IA reconstruction signal, the grey line in Fig. 5a). Fig. 5b shows that the near 9-year cycle in Tmx correlates well with the main mode of the first global SST pattern (Fig. 2a), with signals over the Southern Atlantic, Indian and Pacific Oceans. The interannual modes exhibit a weak effect on the minimum temperature (Fig. 5c) as they account for just 30% of Tmn's variability. Yet, the minimum temperature is related to the ENSO mode with a periodicity close to 3 years (Fig. 5d) with warmer Tmn during El Niño years and cooler Tmn during La Niña events. These results are consistent with Müller et al. (2000) and Rusticucci et al. (2017) who reported that El Niño events intensify warm spells and reduce the number of climatological freezing days while La Niña events increase cold events.

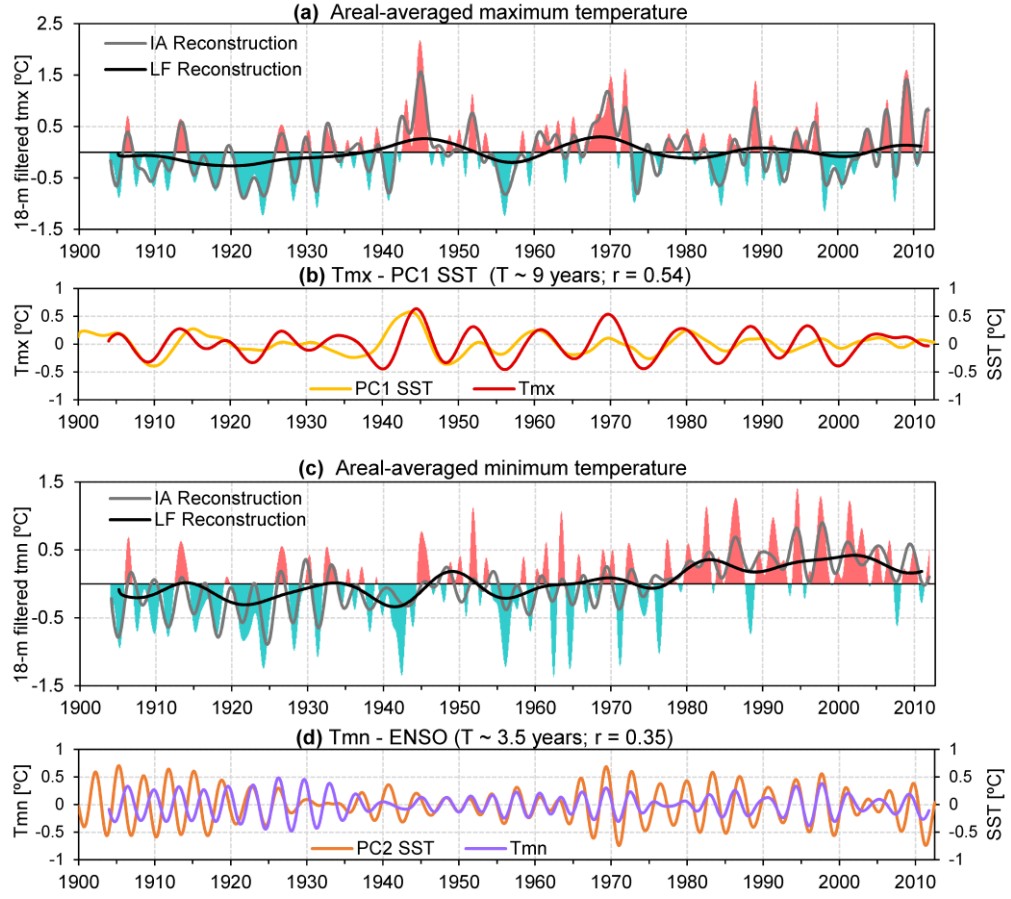

**Figure 5: Panels (a) and (c) present the 18-m low-pass filtered monthly time series of areal-averaged daily maximum and minimum temperature anomalies (in shades) and their partial reconstructions (in lines) using all interannual and low-frequency variability modes described in Table 2. Panels (b) and (d) depict the temporal evolution of the dominant modes of regional temperature and the first two leading patterns of global SST variability. T denotes the dominant period of the leading modes. r indicates the Pearson correlation coefficients between time series.**




## 4.2 Trends and decadal-to-multidecadal oscillations

Figures 6a and 6b show the leading pattern of decadal-to-multidecadal precipitation variability. This pattern presents correlation values mostly larger than 0.7 and maximum values towards the central-western portion of NEA. A noticeable nonlinear trend in the PC1 explains 60% of its variance. The PC1 nonlinear trend shows two distinct periods of the

5    precipitation: a dry one before 1970 and a wet one between 1970 and 2005. The wetting trend has stabilized and reversed starting in the 2000s (in agreement with Seager et al., 2010; Lovino et al., 2014; Saurral et al., 2017). The PC1 also exhibits interdecadal cycles of about 15-20 years (see Table 2). The second leading pattern of precipitation (Figs. 6c and 6d) displays negative correlations on the central-eastern portion of the domain and mostly weak positive correlations to the north and southwest. The PC2 has a near-decadal cycle and a multidecadal oscillation (Table 2) that explain 25% and 58% of the PC2

10   variance respectively. As reported by Seager et al. (2010), Figs. 6b and 6d show that there was a multiyear period of droughts in the 1930s and 1940s coincident with negative anomalies in the low-frequency trend of PC1 and a dry episode of the multidecadal oscillation of PC2. This multiyear drought was called Pampas Dust Bowl (Viglizzo and Franck, 2006) and was caused by a hemispherical symmetric pattern of precipitation anomalies with drought in both North and South America (Seager et al., 2010).

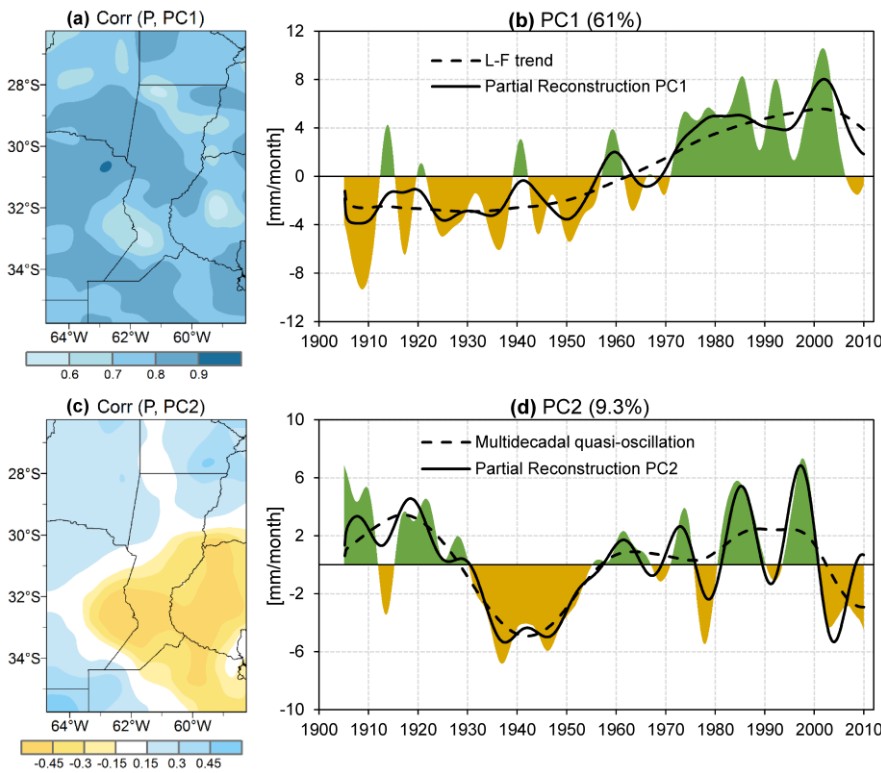

**Figure 6: First two leading patterns of decadal-to-multidecadal precipitation variability characterized by the 120-m low-pass filtered monthly mean precipitation anomalies. Panels (a) and (c) show the spatial distribution of the correlation between PCs (in b and d) and each precipitation time series at a single grid point. Percentage of explained variance are given in brackets. Black lines depict the leading modes of variability (dashed) and the partial reconstruction of each PC using all modes described in Table 2 (continuous).**



Figure 7a shows that the multidecadal oscillation of precipitation (green line) has positive correlations with the PC2 (lf) SST (Interdecadal Pacific Oscillation, section 3), and negative correlation with the PC3 (lf) SST (Atlantic Multidecadal Oscillation, section 3). Thus, consistent with Seager et al. (2010), Mo and Berbery (2011) and Barreiro et al. (2014), warm phases of the

5    Interdecadal Pacific Oscillation and cold phases of the Atlantic Multidecadal Oscillation favour long-period wet anomalies in decadal time scales over NEA. It could be the case of the wet period from the 1970s to 2000s in regional precipitation as seen in Fig. 7a. Conversely, the reversal in the wetting period since the mid-2000s can be explained as a transition to a cold phase of the Pacific Ocean and a warm period of the Atlantic Ocean.

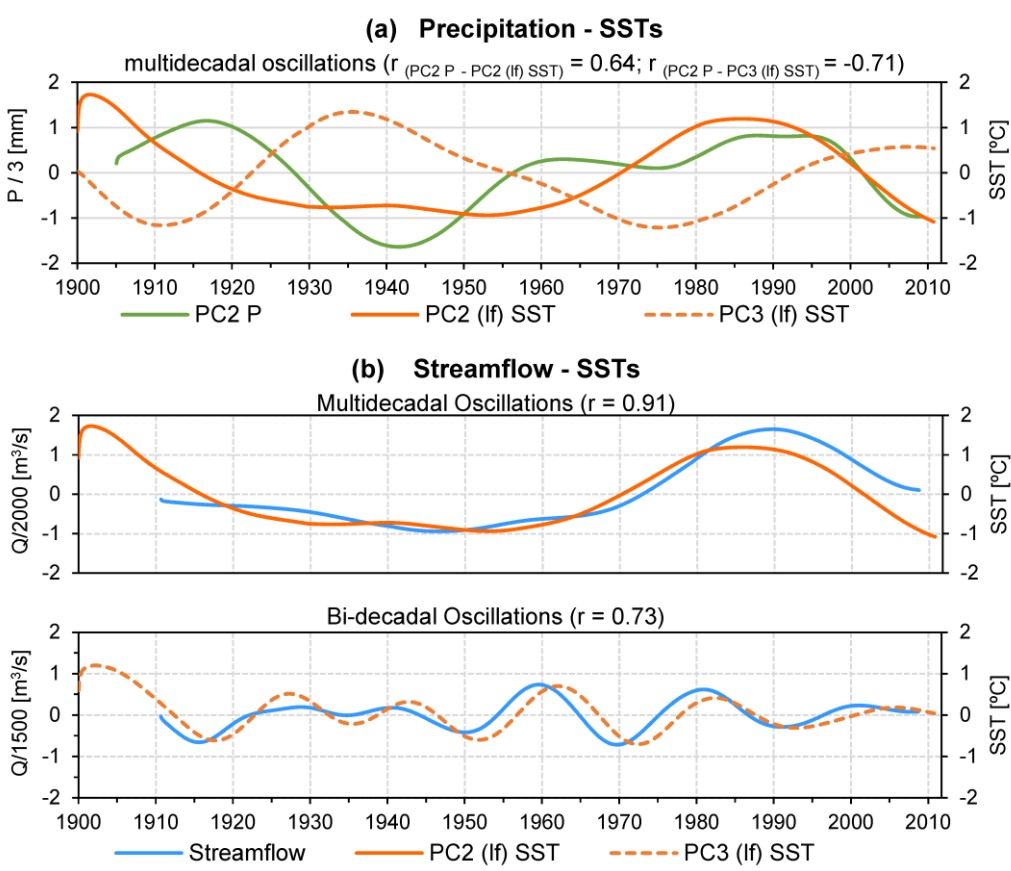

**Figure 7: Time series of the dominant modes of regional hydroclimates variables and the leading patterns of global SSTs at decadal-to-multidecadal time scales. The description of the modes is given in Tables 1 and 2: (a) Multidecadal oscillations in PC2 SST, PC3 SST and PC2 P. (b) Multidecadal Oscillations in streamflow and PC2 (top panel); bi-decadal oscillation in streamflow and PC3 SSTs (bottom panel). r is the Pearson correlation coefficient between series.**

The low-frequency variability of the Paraná River streamflow exhibits a near bi-decadal and a multidecadal oscillation that

15   represent 20% and 60% of the total variance respectively (Table 2). First, in agreement with Labat et al. (2005), Antico et al. (2014), and Antico et al. (2016), the top panel of Fig. 7b shows that the multidecadal oscillation of streamflow strongly correlates (r~0.91) with the low frequency oscillation of the global SST (lf) PC2 related to the interdecadal Pacific variability.



Thus, the warmest phase of the Interdecadal Pacific Oscillation favoured the extraordinary floods between 1980 and 2000 (see dots in Fig. 4a). Second, the bottom panel of Fig. 7b shows that the near bi-decadal mode of streamflow correlates (r~0.73) with a similar mode in PC3 (lf) SST corresponding mainly to the North and South Atlantic (Fig. 2k). Thus, our results suggest that North and South Atlantic SSTs influence the Paraná River streamflow with bi-decadal periodicities. In addition, Robertson

and Mechoso (2000) and Antico et al. (2014) reported that the South Atlantic Convergence Zone also drives the Paraná River flow with 18-yr periodicities.

Maximum and minimum near surface temperatures also present a near bi-decadal oscillation and a nonlinear trend (Table 2). The maximum temperature mostly varies with a bi-decadal frequency (Fig. 5a) that induced larger values around the 1940s, the 1960s, and 2000s. The minimum temperature is mostly dominated by a nonlinear trend (Fig. 5c) that favoured a notable

increase since the 1970s.

## 5 Sectoral impacts of hydroclimatic variability

Hydroclimatic variability at different time scales led to severe sectoral impacts in northeastern Argentina (Barros et al., 2015). As stated by Magrin et al. (2014), southeastern South America presents high vulnerability to extreme events mainly related to intense rainfall events and interannual time scales of hydroclimatic variability (largely influenced by the ENSO). Trends and

decadal-to-multidecadal oscillations have exacerbated some of these extremes, particularly in the wet-period between 1970 and 2005 when the most catastrophic floods occurred. Based on the evidence presented on section 4 and the scientific literature available for southeast South America, the main impacts of hydroclimate variability on regional water resources, agriculture and human settlements are discussed on this section.

### 5.1 Water Resources

Water resources in northeastern Argentina are highly sensitive to hydroclimate variability and its extremes. Intense precipitation events of shorter duration (days to weeks) originated in mesoscale and convective systems are common in NEA (Rasmussen et al., 2016). These events produce localized sudden floods in plain areas that affect water resources management systems, i.e., cause severe complications in urban and rural drainage systems (Bertoni et al., 2004). Persistent events (from seasons to years) affect water supply management since groundwater, streamflow, and reservoir storage reflect the longer-term

precipitation anomalies (Latrubesse and Brea, 2011; Penalba and Rivera, 2016). These events, as discussed above, are mainly driven by extreme ENSO conditions that induce floods or droughts (row 1 of Table 3).

For instance, during the 2008/2009 and 2012 droughts the Salto Grande hydropower plant on the Uruguay River generated only 10% of its energy capacity impacting on the energetic security of Uruguay and Argentina (El Día, 2012). These droughts affected navigability of the Paraná River leading to complications to export grains and transport fuel (La Nación, 2009). The

very low levels of the Paraná and Uruguay rivers hindered water extraction for supply leading to shortages of drinking water and pollution problems (La Nación, 2009).



| Time scale | Key impacts in water resources | Key references |
|---|---|---|
| **Interannual** | Flooding mostly during El Niño years and droughts generally associated with La Niña events affect water management. Difficulties in water supply, hydroelectric power generation and river navigability. | Grimm et al. (2000) Grimm and Tedeschi (2009) Garreaud et al. (2009) Latrubesse and Brea (2011) Barros et al. (2015) Penalba and Rivera (2016) |
| **Interannual to multidecadal** | Severe hydrological droughts, mainly in 1901-1960. | Viglizzo and Frank (2006) Tasso et al. (2011) Lovino et al. (2014) |
| **Decadal to Multidecadal** | Increased precipitation caused increased flows. Frequent regional flooding between 1970 and 2005. Increased groundwater levels favored surface runoff. Extraordinary fluvial floods. | Venencio and García (2011) Latrubesse and Brea (2011) Cavalcanti (2012) Magrin et al. (2014) Barros et al. (2015) |

**Table 3: Impacts of hydroclimate variability and its extremes in water resources. The first column shows the main time scale of variability affecting the study sector.**

Our results show that northeastern Argentina suffered severe droughts that were more frequent before 1960, i.e., during the early 20th century (row 2 on Table 3). During those years, the droughts severely affected regional economies with similar impacts to those discussed above but exacerbated by the scarce technological development at that time (Tasso, 2011).

Decadal-to-multidecadal time scales (including trends) affect the precipitation and streamflow variability (row 3 of Table 3), including increases in precipitation and streamflow after the 1970s as well as frequent wet extreme events between 1970 and 2005. These phenomena raised regional groundwater table levels, reduced the soil infiltration capacity and favoured increased surface runoff (e.g., Venencio and García, 2011; Latrubesse and Brea, 2011). All these changes combined with the significant hydrological variability on interannual time scales favoured extraordinary floods in the main rivers in last decades.

In 1983 and 1998, the extraordinary floods of the Parana and Uruguay River forced the evacuation of hundreds of thousands of people and an estimated cost of over a billion dollars (Barros et al., 2015). During those years, the Paraná River floods affected approximately 40,000 km² for 8 months (Fritschy, 2010; Barros et al., 2015). Moreover, the Salado river—a tributary of the Paraná river—experienced the most catastrophic flood in 2003, causing economic losses of approximately US$ 1000 million and dozens of dead in the city of Santa Fe (ECLAC, 2003).

In the coming decades, it is expected that flooding in the Paraná River occur frequently due to the influence of interannual variability. As noted in section 4b, the "positive" phase of the multidecadal hydrological variability that exacerbated the magnitude of floods during the 1980-2005 period was weakened in the last decade and could even reverse in the coming years. Our results suggest that the change of phase in multidecadal streamflow variability was induced by a reversal in the precipitation-wetting trend and in the multidecadal modes of streamflow. Seager et al. (2010) argued that the reversal of the precipitation trend is influenced by a shifting toward a positive phase of the Atlantic Multidecadal Oscillation (Ting et al.,



2009), which may drive a precipitation decrease in the coming years. Thus, if these conditions remain, the influence of multidecadal variability could attenuate the magnitudes and devastating impacts of the floods compared to those that occurred during the 1980-2005 period.

## 5.2 Agriculture

5   Agricultural activities were favoured by the long-term wet period since 1970, improved technology and increased planted area: a significant increase in annual crops of around 60% occurred in the whole region (Viglizzo et al., 2011). Wetter conditions also led to increases in crop yields (maize and soybean yields increased 9 and 58% respectively; Magrin et al., 2007). The wet period since 1970 as well as economic, social and technological drivers contributed to the expansion of agriculture over regions previously relegated like the Chaco dry forest, toward northwest of the study region (Paruelo, 2005). It leads to a strong process

10  of land use changes: about 80% of the natural forest has been converted to pastures, scrub or cropland (Zak et al., 2008; Viglizzo et al., 2011). According to Nosetto et al. (2008) and Magrin et al. (2014) those land use changes have disrupted natural water and biogeochemical cycles, affecting surface runoff and salinizing the soil.

| Time Scale | Key impacts in agriculture | Key references |
|---|---|---|
| **Decadal to multidecadal** | Wet period 1970-2005 favored an extension of agricultural lands to marginal areas but also land use changes. | Paruelo (2005)<br>Zak et al. (2008)<br>Barros et al. (2008b)<br>Nosetto et al. (2008)<br>Viglizzo et al. (2011)<br>Magrin et al. (2014) |
| **Interannual to multidecadal** | Extensive and frequent flooding since 1970 reduced agricultural and livestock productivity. Large and severe droughts affect agricultural and livestock activities and favor wind erosion. | Herzel et al. (2004)<br>Minetti et al. (2010)<br>Magrin et al. (2014)<br>Cavalcanti et al. (2015)<br>Anderson et al. (2017) |
| **Decadal to multidecadal** | Increased minimum temperature cause wheat and barley yield decreases and possible modification of the maize crop cycle. | Magrin et al. (2009)<br>Maddonni (2012)<br>Fernandez Long et al. (2013)<br>Verón et al. (2015)<br>Müller et al. (2015)<br>García et al. (2015) |

**Table 4: Impacts of hydroclimate variability and its extremes in agriculture. The first column shows the main time scale of variability affecting the study sector.**

15  The combination of interannual and multidecadal hydrological variability led to extreme events (floods and droughts) that affected agricultural and livestock productivity (row 2 of Table 4). According to Magrin et al. (2014), river floods and extensive pluvial flooding, more frequent after 1970, affected riverine and rural areas causing losses or damages in pastures and crops and forcing displacement of cattle. For instance, the Paraná flood in 1998 affected 3.5 million of productive hectares and



caused losses of 750 million U.S. dollars in the agricultural sector of Argentina (Herzel et al., 2004). The extensive pluvial floods of 2016 affected 6 millions of productive hectares in the core crop region of Argentina causing losses for 2750 million dollars including crop damages, animal mortality, pasture and inputs, and the destruction of dairy farm equipment (La Nación, 2016).

Within the dry period (1901-1960), a multiyear drought between 1925 and 1940 extended the "Pampas Dust Bowl" to all of northeastern Argentina (Lovino et al., 2014). Extremely dry conditions produced cattle mortality, crop failure, farmer bankruptcy and rural migration (Viglizzo and Frank, 2006). Even during the wet period (after the 1970s), extended agricultural droughts altered sowing or critical crop growth periods decreasing yields (Minetti et al., 2007; 2010). For instance, the last two severe droughts in 2008/2009 and 2011/2012 caused losses of 8,765 million dollars to the agricultural sector of the core

crop region in Argentina (El Economista, 2016). In this region, the severe drought in 2008 reduced crop yields by 40% and caused the mortality of 700,000 cattle (La Nación, 2008).

The variability and changes in temperature also affect the agricultural and livestock sector (row 3 of Table 4). The increase of minimum temperature results in increased wheat and barley respiration rates and shorter grain filling period diminishing their yields (Magrin et al., 2009; Verón et al., 2015). According to García et al. (2015), wheat and barley yields were reduced under

increased minimum temperatures by 7% without considering technological improvements. The life cycle of maize has been affected by fewer frost events that allow early planting dates (Maddonni, 2012).

### 5.3 Human Settlements

Frequent pluvial and river floods affected urban and rural populations in the Lower Paraná basin (rows 1 and 2 of Table 5). River floods affect mostly vulnerable social sectors such as unplanned urban settlements, usually located in river floodplain

areas, while pluvial floods affect established rural and urban populations spread over large areas of the Pampas.

| Time scale | Key impacts in human settlements | Key references |
|---|---|---|
| **Interannual to multidecadal** | River floods impacted riverside populations. Alteration of settlements, commerce, transport and pressure on urban infrastructure. | ECLAC (2003) Paoli et al. (2004) Barros et al. (2015) |
| **Interannual to multidecadal** | Extensive pluvial floods affected urban and rural settlements. Isolation of rural populations. | Bertoni et al. (2004) Giacosa et al. (2004) Latrubesse and Brea (2014) |
| **Interannual to multidecadal** | Severe droughts can cause scarcity of water and food and less potential for hydropower. | Rivera and Penalba (2013) Magrin et al. (2014) |
| **Decadal to multidecadal** | Increased effects of heat waves. Higher air pollution. Increased energy demand for cooling in summer and decreased energy demand for heating in winter. | Rusticucci (2012) Magrin et al. (2014) Rusticucci et al. (2015) Santágata et al. (2017) |

**Table 5: Impacts of hydroclimate variability and its extremes in human settlements. The first column shows the main time scale of variability affecting the study sector.**



There are many severe cases of reported damage to cities due to river floods. For instance, the Paraná River flood in 1982/1983 inundated 70% of the city of Resistencia (located on the opposite margin of the river from the city of Corrientes) and forced the evacuation of 91,500 people (Herzel et al., 2004). The sewer system, urban drainage, and the power system collapsed and the communication lines disrupted resulting in economic losses of 150 million dollars (Caputo et al., 1985). According to

ECLAC (2003), the Salado River flood in the city of Santa Fe in 2003 forced the evacuation of 120,000 people and affected 20,000 houses. This disaster resulted in severe damage to urban infrastructure estimated at 450 million dollars including drinking water and sanitation, energy and transportation and telecommunications networks. Moreover, the extraordinary river floods also affected rural settlements. For instance, the Paraná River flood of 1998 affected 53,000 families of small farmers, rural workers and indigenous communities (Herzel et al., 2004).

Pluvial floods have frequently affected small settlements in the NEA, but have also led to major disasters in densely populated cities. In 2013, an unprecedented intense rainfall that reached 400 mm in less than 12 hours flooded the city of La Plata, the capital city of the province of Buenos Aires. In the urban area, 3,500 hectares were flooded, more than 80 people died and more than 190,000 people were affected and substantial material damage was produced (UNLP, 2013). The city of Santa Fe suffered a pluvial flood in 2007 that affected a great part of the urban area causing three deaths and forcing the evacuation of

26,000 people (La Nación, 2007).

On the other hand, severe droughts altered food security and water availability in populations located towards the northwest of the region where the access to water is hindered (row 3 of Table 5). For instance, the 2008/2009 drought affected water supply and favoured outbreaks of waterborne diseases in several cities in the northwest of the region (La Nación, 2009). Also, the lack of water forced the migration of rural families to the cities of the region (Infocampo, 2009).

Increased temperature has exacerbated heat waves and elevated the air pollution in several cities of the region (Row 4 of Table 5). According to Magrin et al. (2014), changes in temperature affect energy demand. Warmer conditions altered the energy supply, with decreases in energy consumption for heating during winter, and increases for cooling during summers (Santágata et al., 2017). For instance, Argentina suffered an intense heat wave that killed seven people in December 2013 (BBC news, 2013). The intense heat was exacerbated by power cuts and water shortages, leading to hundreds of cases of heat strokes in

Buenos Aires (Independent, 2013).

## 5 Final remarks

Hydroclimate variability (here described by precipitation, streamflow and near surface temperature) impacts water resources, agriculture, and human settlements in northeastern Argentina, exacerbating social, political, economic, and environmental risks already existing in the region. This study (a) investigated the regional hydroclimate variability on interannual-to-

multidecadal time scales, (b) reviewed and advanced the understanding of its links with global SSTs forcing and (c) discussed the impacts of hydroclimate variability and its extremes on the main productive and socio-economic regional sectors.





The region is affected by hydroclimate variability modes in different time scales, from interannual to multidecadal and likely into trends. Interannual hydrological variability induced frequent floods and droughts that are mainly related to extreme ENSO conditions, often combined with SST forcing in the North Tropical Atlantic Ocean and the South Atlantic Convergence Zone. These extreme events altered the management of surface and groundwater resources. Frequent floods affected agricultural and
livestock productivity in riverine and rural areas of the Pampas causing damages to pastures and crops and forcing displacement of cattle. Human settlements were also affected, including impacts on trade, transportation, and infrastructure of urban and rural populations. Conversely, agricultural droughts disrupted seeding or critical periods of crops and hydrological droughts impacted water supply for cattle while favouring soil erosion. The most severe hydrological droughts affected directly the population causing water and food scarcity and reducing the potential for hydropower generation.
On decadal-to-multidecadal time scales, northeastern Argentina experienced a transition from dry and cooler to wet and warmer decades since the 1970s. The wet period (after 1970) favoured agriculture over regions previously relegated and also led to land use changes. The combined effect of increased precipitation and land use changes resulted in raised groundwater table levels, reduced the soil infiltration capacity and increased surface runoff leading to extraordinary floods. Temperature variability and changes on low-frequency time scales affected productive sectors and urban settlements. Increases in minimum
temperature reduced wheat yields and altered maize life-cycles. Changes in maximum and minimum temperatures exacerbated the effects of urban heat islands and affected energy demand in several cities of the region.

The combination of interannual and multidecadal hydroclimatic variability intensified extreme events in the past, leading to the most severe droughts in the early 20th century and largest floods after 1970. Since the mid-2000s, northeastern Argentina has been showing signs of a reversal in the wetting period on multidecadal time scales. If these conditions continue into the
coming decades, more severe droughts and weaker floods might be expected. Considering the relevance of the multidecadal variability in the formation of extreme events, this information should be monitored and incorporated into regional decision systems to improve planning by water managers and agricultural stakeholders.

**Acknowledgements**

This research was carried out with the support of Projects CRN3035 of the Inter-American Institute for Global Change
Research (IAI), which is supported by the US National Science Foundation. UNL Project C.A.I. + D. 2011 35/180 are also acknowledged. We greatly appreciate the support of Ernesto H. Berbery in the preparation of this manuscript.

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
