# Peer review of "Interannual-to-multidecadal Hydroclimate Variability and its Sectoral Impacts in northeastern Argentina"

_Hydrology and Earth System Sciences, 2018_

## Referee Comment (RC1) · Anonymous Referee #1 · 14 Mar 2018

General comments: The manuscript examines the interannual and multidecadal scales of global SSTs and its relation to precipitation, river flow and maximum and minimum temperature over Northeast Argentina. The authors analyzed and correlated these hydroclimate variables by applying low-pass Lanczos filter, EOFs and SSAs. To conclude their study, the authors analyzed the impacts of hydroclimate variablity and trends on the water, agriculture, and human sectors.

The manuscript is well-written and contributes substantially to the scientific progress within the scope of Hydrology and Earth System Sciences by utilizing a different filtering method compared to the usual running-mean method in analyzing the climatic vari-

ables. The authors used observational and reanalysis datasets including river gauge station datasets. However, the authors were not able to put forward what is something new in their study. I would recommend the authors to highlight the novelty of their work. For example, is the transition from dry and cooler climate to wet and warmer decades in northeastern Argentina novel? Also, the Pearson correlation coefficient method was also used to correlate SST anomalies with precipitation, stream flows and temperature. But, there is no significant tests employed. How significant are these correlations? In addition, some specific comments are mentioned below for consideration.

I recommend the manuscript as accepted with minor revision.

Specific comments:

Abstract: L.15: transition from what climate to wetter and warmer climate?

P.3 Figure 1a. bold black contours and blue polygon Figure 1b. sub-basins are black contours, rivers and tributaries in blue contours

P.4 L. 25. Lanczos filter, how many weights used? Is the use of Lanczos filter more useful compared to the running-mean method?

P.6 Figure 2. missing SST in 120-month low-pass filtered SST label and wrong label for Panel j. What do you mean with partial reconstructions?

P.18 Table 5. Higher concentrations of air pollution

———————————————

---

## Referee Comment (RC2) · Anonymous Referee #2 · 15 Mar 2018

General comments:

The manuscript assesses the joint interannual to multidecadal variability of precipitation, river flow and maximum and minimum temperatures over northeastern Argentina representing a contribution to scientific progress within the scope of Hydrology and Earth System Sciences. Overall, the manuscript is well written and presented in a clear and well structured way. The authors perform an exhaustive review of literature addressing the sectoral impacts of hydroclimate variability and trends including a comprehensive discussion of these impacts at different timescales of interest. Nevertheless I consider the results are not able to show robustly what is the novelty of the present

study. In my opinion the results presented in this research are not well balanced compared to the review the authors perform based on the previous studies in the topic of interest. Some examples of results which are not novel are detailed in the specific comments. Authors should definitely put a major emphasis on what they consider to be new in their study which has not been performed or updated in the existing literature.

I recommend to reconsider the manuscript after major revision.

Specific comments:

-P.2, L.3-5. Authors should try to update references in relation to studies in the region. What has been done since 2010 regarding the understanding of regional hydroclimate and its sectoral impacts to increase the resilience of the affected populations in Argentina and SESA?

-P.2, L.32-33. Authors state they will "assess the impacts of hydroclimate variability and trends on water resources, agriculture and human settlements". Based on Section 5, I consider this is more a review of the existing literature in relation with sectoral impacts (with a comprehensive and appropriate discussion of them at the different timescales) rather than a concrete "assessment" of the results.

-P. 9, L.6-7. "Our results show that all the main frequencies of the ENSO are also replicated in the regional precipitation". I suggest to check for references since these kind of research have been well explored in previous studies and it is not novel.

-P.9, L. 11-25. I found the description of results based on Fig 3b and Fig 3c not new. I suggest to highlight the novelty in these results which has not been discussed by other studies.

-P.11, L.10-17. "These results suggest that all extraordinary floods of Paraná River occurred during strong or very strong El Niño events". Same as above comment but with Figure 4.

Technical corrections:

[Figure]

-Table 2. Include units of the leading frequencies in the table caption. In the text authors refer to period (units: years). Please check to be consistent.

-P.12, L.2. Please revise the exhibited Tmx frequencies which you mention in the text to be close to 3, but results indicate to be 2.4.

---

## Author Comment (AC1) · 28 Mar 2018

**Responses to Reviewers**

We are grateful to the anonymous reviewers whose constructive comments will help to improve the manuscript. We provide an integrated response regarding the novelty of our study, which has been observed by both reviewers (see *section 1: general comment of the authors*). Then we respond to the general and specific comments of each reviewer (see *section 2: detailed responses to reviewers*). The response to the reviewers is structured: (1) comments from reviewers (shaded), (2) author's response immediately below, (3) author's changes in manuscript.

**1. General comment of the authors**

This study offers a joint view of the main hydroclimatic variables at different time scales in northeastern Argentina. To our knowledge, there are no previous studies providing accurate and integrated information of several hydroclimate variables that can be useful for decision making in different sectors. As both reviewers correctly noted, some results regarding hydroclimate variability that we have obtained here had been reported in previous studies for individual variables (e.g., precipitation or streamflow). These results have been compared with those previous studies and they have been carefully referenced. However, this paper also provides novel knowledge that complement the state of the art of regional hydroclimate variability. Some of the novel findings regarding hydroclimate variability and its links with SSTs patterns are now particularly highlighted to reinforce the originality of our paper, including:

- *The increase in the amplitude of 2.5-to-4 year periods in precipitation that favored extreme events after 2000*: We found that interannual precipitation variability with frequencies between 2.5 and 4 years contribute to exacerbate extreme events as of the 2000s, even during moderate extreme phases of the ENSO. This difference could be explained by an increase in heavy rainfall resulting from greater atmospheric instability and water vapour content (see e.g. Re and Barros, 2009; Penalba and Robledo, 2010).

- *The contribution of NEA precipitation rates to the Low Paraná river streamflow with periodicities of near 2 years*: We reported that a near biannual oscillation is noted in Timbúes gauging station (representative of the Lower Paraná Basin) but not in Corrientes gauging station (representative of the Mid-Upper Paraná Basin). Although precipitation rates over NEA contribute with a small percentage to the

total Paraná River flow, this finding can provide valuable information for early warning systems in periods of low flow contribution in the middle and upper reaches of the river.

- Regarding *temperature*, this study reported two interesting links between regional temperature and global SST patterns. *Maximum temperature time series correlates well with SSTs variability over the Southern Atlantic, Indian and Pacific Oceans with a 9-year frequency. Minimum temperature relates with ENSO with a main frequency close to 3 years* [note that it was known that the region experiences warmer minimum temperature during El Niño years and cooler minimum temperature during La Niña events (Müller et al., 2000 and Rusticucci et al., 2017) but not the frequency of this relationship].

- On decadal-to-multidecadal time scales, in addition to the known incidence of the Interdecadal Pacific Oscillation and the Atlantic Multidecadal Oscillation that favour long-period anomalies over NEA (Seager et al., 2010; Barreiro et al., 2014), this study found that *the Paraná River streamflow is influenced by North and South Atlantic SSTs with bi-decadal periodicities*.

These novel findings, now emphasized in the results sections, are also highlighted in the abstract and in section 5 "Final remarks".

Furthermore, this research provides valuable information on the sectoral impacts at different time scales of variability that can support actions and policies seeking to increase the resilience of the regional population to cope with extreme events. The discussion related to sectoral impacts include the following topics:

- Interannual hydrological variability induced frequent floods that affected agricultural and livestock productivity, as well as trade, transportation and infrastructure on urban and rural populations. Conversely, droughts disrupted crop seeding and water supply for cattle and population.

- On decadal-to-multidecadal time scales, the study region experienced a transition from dry and cooler to wet and warmer decades since the 1970s favouring agriculture expansion. It derived in changes in the soil water budget that lead to extraordinary floods. Increases in minimum temperature reduced wheat yields and altered maize life-cycles. Changes in maximum and minimum temperatures exacerbated the effects of urban heat islands and affected energy demand in several cities of the region.

- Since the mid-2000s, northeastern Argentina has been showing signs of a reversal in the wetting period on multidecadal time scales. If these conditions continue into the coming decades, more severe droughts and weaker floods might be expected.

Although the information related to sectoral impacts include some results reported by other authors, this paper provides an integrated and comprehensive discussion relating the hydroclimate variability at different time-scales that affect different sectors (including indicators such as the affected area, the number of people affected, economic losses, among others). The main objective of this part of the work is to improve the understanding of these sectors about how extreme events of hydroclimate variability alter their normal conditions and how do they impact population.

Cited References:

Barreiro, M., Diaz, N. and Renom, M.: Role of the global oceans and land–atmosphere interaction on summertime interdecadal variability over northern Argentina, Climate Dyn., 42, 1733-1753, doi: 10.1007/s00382-014-2088-6, 2014.

Müller, G., Nuñez, M. and Seluchi, M.: Relationship between ENSO cycles and frost events within the Pampa Húmeda region, Int. J. Climatol., 20, 1619–1637, doi:10.1002/1097-0088(20001115)20:13<1619::AID-JOC552>3.0.CO;2-F, 2000.

Penalba, O. C. and Robledo, F. A.: Spatial and temporal variability of the frequency of extreme daily rainfall regime in the La Plata Basin during the 20th century, Climatic Change, 98, 531-550, doi:10.1007/s10584-009-9744-6, 2010.

Re, M. and Barros, V. R.: Extreme rainfalls in SE South America, Climatic Change, 96, 119-136, doi:10.1007/s10584-009-9619-x, 2009.

Rusticucci, M., Barrucand, M. and Collazo, S.: Temperature extremes in the Argentina central region and their monthly relationship with the mean circulation and ENSO phases. Int. J. Climatol., 37: 3003–3017, doi:10.1002/joc.4895, 2017.

Seager, R., Naik, N., Baethgen, W., Robertson, A., Kushnir, Y., Nakamura, J. and Jurburg, S.: Tropical Oceanic Causes of Interannual to Multidecadal Precipitation Variability in Southeast South America over the Past Century, J. Climate, 23, 5517–5539, doi: 10.1175/2010JCLI3578.1, 2010.

**2. Detailed responses to reviewers**

**Reviewer 1**

**General comments:**

The manuscript examines the interannual and multidecadal scales of global SSTs and its relation to precipitation, river flow and maximum and minimum temperature over Northeast Argentina. The authors analyzed and correlated these hydroclimate variables by applying low-pass Lanczos filter, EOFs and SSAs. To conclude their study, the authors analyzed the impacts of hydroclimate variability and trends on the water, agriculture, and human sectors.

The manuscript is well-written and contributes substantially to the scientific progress within the scope of Hydrology and Earth System Sciences by utilizing a different filtering method compared to the usual running-mean method in analyzing the climatic variables. The authors used observational and reanalysis datasets including river gauge station datasets. However, the authors were not able to put forward what is something new in their study. I would recommend the authors to highlight the novelty of their work. For example, is the transition from dry and cooler climate to wet and warmer decades in northeastern Argentina novel? Also, the Pearson correlation coefficient method was also used to correlate SST anomalies with precipitation, stream flows and temperature. But, there is no significant tests employed. How significant are these correlations? In addition, some specific comments are mentioned below for consideration.

I recommend the manuscript as accepted with minor revision.

We have highlighted the novel findings of our study, as clarified in the "general comment of the authors" above (see section 1). Regarding the question, the transition from dry and cooler climate to wet and warmer decades is not novel at all, as it was partially reported by other authors (e.g., Seager et al., 2010; Barreiro et al., 2014, Jaques-Coper and Garreaud, 2015). However, our results reinforce the state of knowledge and evaluate the sectoral impacts of the hydroclimatic trends. For example, our results show jointly that warm phases of the Interdecadal Pacific Oscillation and cold phases of the Atlantic Multidecadal Oscillation favour long-period wet anomalies in decadal time scales over NEA and exacerbate the Paraná River floods.

As suggested by the reviewer, we have incorporated a significance test for Pearson correlations (r). Significance levels for r are estimated by combining 2000 Monte Carlo

iterations considering autocorrelation by resampling in the frequency domain (Macias-Fauria et al., 2012). Significance level are now provided with the correlation values in each figure. The method employed to test significance is mentioned in the figure captions.

Cited references:

Barreiro, M., Diaz, N. and Renom, M.: Role of the global oceans and land–atmosphere interaction on summertime interdecadal variability over northern Argentina, Climate Dyn., 42, 1733-1753, doi: 10.1007/s00382-014-2088-6, 2014.

Jacques-Coper, M. and Garreaud, R. D.: Characterization of the 1970s climate shift in South America, Int. J. Climatol., 35, 2164–2179, doi:10.1002/joc.4120, 2015.

Macias-Fauria, M., Grinsted, A., Helama, S., and Holopainen, J.: Persistence matters: Estimation of the statistical significance of paleoclimatic reconstruction statistics from autocorrelated time series, Dendrochronologia, 30, 179–187, doi:10.1016/j.dendro.2011.08.003, 2012.

Seager, R., Naik, N., Baethgen, W., Robertson, A., Kushnir, Y., Nakamura, J. and Jurburg, S.: Tropical Oceanic Causes of Interannual to Multidecadal Precipitation Variability in Southeast South America over the Past Century, J. Climate, 23, 5517–5539, doi: 10.1175/2010JCLI3578.1, 2010.

**Specific comments:**

Comment 1: Abstract L.15: transition from what climate to wetter and warmer climate?

The statement has been rewritten.

*Change in the manuscript:*

The statement in the abstract now states: "Interdecadal variability is characterized by low frequency trends and multidecadal oscillations that have induced a transition from dry and cooler climate to wet and warmer decades starting in the mid-twentieth century"

Comment 2: P.3 Figure 1a. bold black contours and blue polygon Figure 1b. sub-basins are black contours, rivers and tributaries in blue contours

The caption of Figure 1 has been rewritten as suggested by the reviewer

*Change in the manuscript:*

The caption of Figure 1 now states "Figure 1: (a) South America and the domains of the La Plata Basin (bold black contour) and southeastern South America (blue polygon). (b) Topography map of La Plata Basin and its principal sub-basins (black contours), Paraguay (PAY), Mid-Upper Paraná (MUP), Lower Paraná (LOP) and Uruguay (URU). The rivers and their tributaries are plotted in blue contours. Gauging stations of the Parana River streamflow in Corrientes ($Q_{co}$) and Timbúes ($Q_{ti}$) are highlighted. The study area in northeastern Argentina is highlighted with a red rectangle."

Comment 3: P.4 L. 25. Lanczos filter, how many weights used? Is the use of Lanczos filter more useful compared to the running-mean method?

We have used 36 weights in the low-pass Lanczos filters. This point is now clarified in the manuscript. The Lanczos filter has advantages compared to the running-mean method that are now briefly discussed.

*Change in the manuscript:*

The first paragraph of section 2.3 "Statistical approach" now explains: "The present analysis focuses on precipitation, streamflow and temperature (minimum and maximum) of the NEA, considering SSTs as the main external forcing. For these four variables, monthly mean anomalies are computed by removing the corresponding mean annual cycle. The variables are also filtered applying low-pass Lanczos filters (Duchon, 1979) with 36 weights and cut-off periods at 18 and 120 months to emphasize the interannual and low frequency behaviour respectively. Lanczos filter is quite simple and yields better response than other filters such as running-mean filter (Duchon and Hale, 2012); for example, Lanczos filter reduces the amplitude of the Gibbs oscillation and allows controlling the cut-off frequencies independently of the number of weights (Duchon, 1979; Navarra, 1999)."

Cited references:

Duchon, C. E.: Lanczos filtering in one and two dimensions, J. Appl. Meteor., 18, 1016-1022, doi: 10.1175/1520-0450(1979)018<1016:LFIOAT>2.0.CO;2, 1979.

Duchon, C., Hare, R.: Chapter 3: Filtering Data, in: Time Series Analysis in Meteorology and Climatology: An Introduction, John Wiley & Sons, New York, United States, 143-182, 2012.

Navarra, A.: Beyond El Niño: decadal and interdecadal climate variability. Springer-Verlag, Berlin, Germany, 374 pp, 1999.

Comment 4: P.6 Figure 2. missing SST in 120-month low-pass filtered SST label and wrong label for Panel j. What do you mean with partial reconstructions?

Figure 2 has been corrected. The original time series are partially reconstructed using the significant components detected with SSA. Now it is explained in the caption of Figure 2.

*Change in the manuscript:*

The caption of Figure 2 now states "Figure 2: First three leading global pattern of 18- and 120-month low-pass filtered monthly mean SST between jan-1901 and sep-2012. Patterns are described by the spatial loadings (EOFs) and their associated PCs. The percentage of variance explained by each pattern is given in brackets. The original time series are partially reconstructed using the significant components detected with SSA. Partial reconstructions at interannual time scales are plotted in PCs of the left column (IA Reconstruction). Partial reconstructions at low-frequency time scales (LF Reconstruction) are plotted in PCs (lf) of the right column. The partial reconstructions include all variability modes at each time scale shown in Table 1. Panels (j) and (l) also show the multidecadal oscillations".

Comment 5: P.18 Table 5. Higher concentrations of air pollution.

Agreed. This change has been made in Table 5.

**Reviewer 2**

**General comments:**

The manuscript assesses the joint interannual to multidecadal variability of precipitation, river flow and maximum and minimum temperatures over northeastern Argentina representing a contribution to scientific progress within the scope of Hydrology and Earth System Sciences. Overall, the manuscript is well written and presented in a clear and well structured way. The authors perform an exhaustive review of literature addressing the sectoral impacts of hydroclimate variability and trends including a comprehensive discussion of these impacts at different timescales of interest. Nevertheless I consider the results are not able to show robustly what is the novelty of the present study. In my opinion the results presented in this research are not well balanced compared to the review the authors perform based on the previous studies in the topic of interest. Some examples of results which are not novel are detailed in the specific comments. Authors should definitely put a major emphasis on what they consider to be new in their study which has not been performed or updated in the existing literature.

I recommend to reconsider the manuscript after major revision.

As suggested by the reviewer, we have highlighted the novel findings of our study, as clarified in the "general comment of the authors" above. The results sections have been rewritten emphasizing the new results of this study and summarizing those that are less novel. We have incorporated the novel results in the abstract and in section 5 "Final remarks". Major changes to the manuscript have been made based on the specific comments that are detailed below.

**Specific comments:**

Comment 1: P.2, L.3-5. Authors should try to update references in relation to studies in the region. What has been done since 2010 regarding the understanding of regional hydroclimate and its sectoral impacts to increase the resilience of the affected populations in Argentina and SESA?

As suggested by the reviewer, the paragraph regarding impacts of hydroclimate variability has been partially rewritten and the references have been updated.

*Change in the manuscript:*

The second paragraph of the introduction now states: "The impacts of hydroclimate variability are more evident in regions where population and the productive sectors are vulnerable to climate hazards. Southeastern South America (SESA) is one such region as documented by Magrin et al. (2014). Frequent flooding impacted large populated areas over SESA (Andrade and Scarpatti, 2007; Barros et al., 2008a; Latrubesse and Brea, 2012). The extraordinary flood along the Paraná River in 1983 produced economic losses of more than $1 billion and forced the evacuation of at least 100,000 people (Krepper and Zucarelli, 2010). The extended and persistent drought of 2008/2009 caused losses of about 40% of grain production in Argentina and greatly affected the hydroelectric sector over SESA (Bidegain, 2009). The severe drought in 2011/2012 caused economic losses of $2.5 in crop production of soybean and corn in Argentina (Webber, 2012). In this context, studies in recent years have been focusing on understanding the regional hydroclimate and its sectoral impacts to increase the resilience of the affected populations by providing adequate information that will facilitate decision-making processes (e.g., Magrin et al., 2014 and references therein; Barros et al., 2015 and references therein; Hernández et al., 2015; Mussetta et al., 2016).

Barros, V. R., Boninsegna, J. A., Camilloni, I. A., Chidiak, M., Magrín, G. O. and Rusticucci, M.: Climate change in Argentina: trends, projections, impacts and adaptation, WIREs Clim Change, 6: 151–169. doi:10.1002/wcc.316, 2015.

Hernandez, V., V. Moron, F.F. Riglos, and E. Muzi,: Confronting Farmers' Perceptions of Climatic Vulnerability with Observed Relationships between Yields and Climate Variability in Central Argentina. Wea. Climate Soc., 7, 39–59, https://doi.org/10.1175/WCAS-D-13-00062.1, 2015.

Magrin, G. O., Marengo, J. A., Boulanger, J.-P., Buckeridge, M. S., Castellanos, E., Poveda, G., Scarano, F. R. and Vicuña, S.: Central and South America, in: Climate Change 2014: Impacts, Adaptation, and Vulnerability. Part B: Regional Aspects. Contribution of Working Group II to the Fifth Assessment Report of the Intergovernmental Panel on Climate Change, Barros, V. et al eds., Cambridge University Press, Cambridge, United Kingdom and New York, NY, USA, pp. 1499-1566, 2014.

Mussetta P., Turbay S., Fletcher A.J.: Adaptive Strategies Building Resilience to Climate Variability in Argentina, Canada and Colombia. In: Leal Filho W., Musa H., Cavan G., O'Hare P., Seixas J. (eds) Climate Change Adaptation, Resilience and Hazards. Climate Change Management. Springer, Cham, 2016.

Webber, J. (Financial Times): Argentina's drought: counting the costs, available at: https://www.ft.com/content/f7fd1da9-9848-39bc-9e15-168d0bd14dd7, last access 27 March 2018, 2012.

Comment 2: P.2, L.32-33. Authors state they will "assess the impacts of hydroclimate variability and trends on water resources, agriculture and human settlements". Based on Section 5, I consider this is more a review of the existing literature in relation with sectoral impacts (with a comprehensive and appropriate discussion of them at the different timescales) rather than a concrete "assessment" of the results.

Thanks for making this important point. Effectively, "assess" is not appropriate for this objective. Based on the evidence regarding hydroclimate variability at different time scales provided in section 4 and the scientific literature available, we provide an integrated and comprehensive discussion of the sectoral impacts of this regional variability. The discussion is supported by some key cases of extremes of climate variability. We analyse how these extremes affected a given region and how do they impacted population (including indicators such as the affected area, the number of people affected, economic losses, among others). Thus, the statement of the third purpose of the study has been rewritten.

*Change in the manuscript:*

The sentence now states: "Lastly, to discuss the impacts of trends and the different time scales of hydroclimate variability on water resources, agriculture and human settlements".

Comment 3: P. 9, L.6-7. "Our results show that all the main frequencies of the ENSO are also replicated in the regional precipitation". I suggest to check for references since these kind of research have been well explored in previous studies and it is not novel.

The statement has been rewritten and referenced as suggested by the reviewer.

*Change in the manuscript:*

The statement now indicates: "Fig. 3c shows that all the main frequencies of ENSO are also replicated in the regional precipitation, in agreement with Penalba and Vargas (2004) and Garreaud et al. (2009)"

Garreaud, R. D., Vuille, M., Compagnucci, R. and Marengo, J.: Present-day South American climate, Paleog. Paleoclim. Paleoecol., 281, 180–195, doi:10.1016/j.palaeo.2007.10.032, 2009.

Penalba, O. C. and Vargas, W. M.: Interdecadal and interannual variations of annual and extreme precipitation over central-northeastern Argentina. Int. J. Climatol., 24: 1565-1580. doi:10.1002/joc.1069, 2004.

Comment 4: P.9, L. 11-25. I found the description of results based on Fig 3b and Fig 3c not new. I suggest to highlight the novelty in these results which has not been discussed by other studies.

As suggested by the reviewer, we have highlighted the novel findings based on Fig. 3. The novelty from Fig. 3b and 3c is that the shorter periods of interannual precipitation variability between 2.5 and 4 years contribute to exacerbate extreme events as of the 2000s even during moderate extreme phases of the ENSO. This result is now better explained and discussed.

 *Change in the manuscript:*

The paragraph that discusses Figs. 3b and 3c now states "Interannual variability of precipitation has a close relation with extreme events (see the dots in Fig. 3b). In agreement with Grimm et al. (2000) and Grimm and Tedeschi (2009), Figs. 3b and 3c show that some of the largest extreme precipitation events occurred during extreme ENSO years, including wet extreme events in 1914, 1972-1973 and 1997-1998 and severe droughts in 1916-1917 and 1988-1989, when SSTs and precipitation cycles were mostly in phase and with high amplitudes. However, the ENSO phenomenon by itself is not enough to explain the intensity of the droughts and wet events. First, extreme precipitation anomalies were recorded during the 2002-2003 moderate El Niño event, and conversely, the 2008-2009 intense drought occurred under moderate La Niña conditions. Studies have shown that other ocean forcing, as the SST over the North Tropical Atlantic Ocean or the South Atlantic Convergence Zone can combine with ENSO to intensify the wet or dry signals in precipitation, leading to extreme events (Robertson and Mechoso, 2000; Seager et al., 2010; Mo and Berbery, 2011). Second, the precipitation for the 2.5- to 4-yr periods exhibits a large increase in amplitude after 2000 as noted in mid-panels in Fig. 3c. Yet, ENSO-like SST cycles were weakened for the same period. Thus, our results suggest that interannual precipitation variability with frequencies between 2.5 and 4 years contribute to exacerbate extreme events as of the 2000s, even during moderate extreme phases of the ENSO. This difference could be

explained by an increase in heavy rainfall resulting from greater atmospheric instability and water vapour content (see e.g. Re and Barros, 2009; Penalba and Robledo, 2010). Moreover, these intense precipitation events are strongly influenced by cycles between 2 and 4 years (Lovino et al., 2018)."

Comment 5: P.11, L.10-17. "These results suggest that all extraordinary floods of Paraná River occurred during strong or very strong El Niño events". Same as above comment but with Figure 4.

Agreed. Our novel findings include: (a) the ENSO-related 6-years cycle is the largest contributor to the formation of extraordinary floods, and (b) the local contribution of the NEA with a periodicity of 2.5 years intensified the highest floods during the studied period. The paragraph that discuss Paraná River floods has been rewritten highlighting our novel findings.

*Change in the manuscript:*

"Floods in the Paraná River are highly influenced by the interannual variability (see dots in Figs. 4a). In agreement with Antico et al. (2016), the largest floods that occurred in 1966, 1983, 1992 and 1998 (dots in Fig. 4a) are echoed with peaks for the different interannual modes of streamflow and ENSO-like SSTs (Fig. 4b). Thus, consistent with Camilloni and Barros (2003) and Antico et al. (2014), Fig. 4 shows that all the extraordinary floods of the Paraná River have occurred during strong or very strong El Niño events. Remarkably, the top panel of Fig. 4b indicate that 6-years cycle is the largest contributor to the formation of extraordinary floods. As state above, the upper Paraná basin provides the largest amount of water to the Paraná River flow. Precipitation rates over NEA account for about 5% of the total Paraná River flow (Giacosa et al., 2000). Thus, the lower Paraná floods are a direct consequence of excess precipitation in the Upper Paraná Basin that has a closely link with extreme El Niño events (e.g., Camilloni and Barros, 2003; Krepper et al., 2008). However, the bottom panel of Fig. 4b suggests that the local contribution of the NEA with a periodicity of 2.5 years intensified the highest floods during the studied period."

**Technical Corrections**

Table 2. Include units of the leading frequencies in the table caption. In the text authors refer to period (units: years). Please check to be consistent

Done.

*Change in the manuscript:*

The caption of Table 2 now states: Table 2. Leading frequencies (in years) of regional precipitation patterns, monthly mean streamflow of the Paraná River, and area-averaged maximum and minimum temperature (Tmx and Tmn). Precipitation is represented by the two leading patterns of the PCA for both 18- and 120-month filtered fields. All streamflow modes were noted in Corrientes and Timbúes gauging stations, except the near biannual cycle (*) that is only shown in Timbúes. The percentage of variance explained by each mode is given in parenthesis.

P.12, L.2. Please revise the exhibited Tmx frequencies which you mention in the text to be close to 3, but results indicate to be 2.4.

Done.

*Change in the manuscript:*

The statement has been rewritten "Table 2 shows that Tmx exhibit frequencies between 2.4 and 3.5 years and of roughly 9 years while Tmn presents cycles between 3 and 6.5 years within the higher interannual band".

---

## Author Response (AR2)

**Response to the Editor**

We are grateful to the editor whose constructive comments and suggestions helped to improve the manuscript. As suggested by the editor, we have highlighted the novelty of our study. To achieve this goal, the abstract and the introduction have been partially rewritten. The abstract has been shortened and improved. The introduction has also been shortened and rewritten more directly. It now clearly explains the contribution of the study in the field of hydroclimatology. The final paragraph of the introduction highlights the plans to meet the proposed objectives and establishes what will be the novel findings and the contributions of the work to the state of the art.

[revised manuscript text omitted]